# Blockchain-Based Data Access Control and Key Agreement System in IoT Environment

**DOI:** 10.3390/s23115173

**Published:** 2023-05-29

**Authors:** JoonYoung Lee, MyeongHyun Kim, KiSung Park, SungKee Noh, Abhishek Bisht, Ashok Kumar Das, Youngho Park

**Affiliations:** 1School of Electronic and Electrical Engineering, Kyungpook National University, Daegu 41566, Republic of Korea; harry250@knu.ac.kr (J.L.); kimmyeong123@knu.ac.kr (M.K.); 2Decentralized Network Research Section, Electronics and Telecommunications Research Institute, Daejeon 34129, Republic of Korea; ks.park@etri.re.kr (K.P.); sknoh@etri.re.kr (S.N.); 3Center for Security, Theory and Algorithmic Research, International Institute of Information Technology, Hyderabad 500 032, India; abhishek.bisht@research.iiit.ac.in (A.B.); or ashok.das@iiit.ac.in (A.K.D.); 4School of Electronics Engineering, Kyungpook National University, Daegu 41566, Republic of Korea

**Keywords:** IoT data, CP-ABE, data validation, data nonrepudiation, data accountability, security, authentication

## Abstract

Recently, with the increasing application of the Internet of Things (IoT), various IoT environments such as smart factories, smart homes, and smart grids are being generated. In the IoT environment, a lot of data are generated in real time, and the generated IoT data can be used as source data for various services such as artificial intelligence, remote medical care, and finance, and can also be used for purposes such as electricity bill generation. Therefore, data access control is required to grant access rights to various data users in the IoT environment who need such IoT data. In addition, IoT data contain sensitive information such as personal information, so privacy protection is also essential. Ciphertext-policy attribute-based encryption (CP-ABE) technology has been utilized to address these requirements. Furthermore, system structures applying blockchains with CP-ABE are being studied to prevent bottlenecks and single failures of cloud servers, as well as to support data auditing. However, these systems do not stipulate authentication and key agreement to ensure the security of the data transmission process and data outsourcing. Accordingly, we propose a data access control and key agreement scheme using CP-ABE to ensure data security in a blockchain-based system. In addition, we propose a system that can provide data nonrepudiation, data accountability, and data verification functions by utilizing blockchains. Both formal and informal security verifications are performed to demonstrate the security of the proposed system. We also compare the security, functional aspects, and computational and communication costs of previous systems. Furthermore, we perform cryptographic calculations to analyze the system in practical terms. As a result, our proposed protocol is safer against attacks such as guessing attacks and tracing attacks than other protocols, and can provide mutual authentication and key agreement functions. In addition, the proposed protocol is more efficient than other protocols, so it can be applied to practical IoT environments.

## 1. Introduction

As IoT devices are deployed in various environments such as houses, farms, factories, and grids, the development and spread of smart cities such as smart homes, smart factories, and smart grids continues. As the amount of data generated and collected by IoT devices increases exponentially, it is predicted that the total amount of data generated annually by 2024 will reach 149 ZB [1]. IoT data are used as source data for services related to finance, medical care, artificial intelligence, and electricity bills.

Data access control technology that can provide IoT data to data users (e.g., managers of smart grids and financial institutions) in an appropriate service environment is required to utilize IoT data as source data for various services. To efficiently utilize IoT data and provide them to data users, the gateway collects IoT data, outsources them to a cloud server, and manages the data through the cloud [2,3]. However, the generated IoT data contain sensitive information such as user personal information, so privacy cannot be guaranteed if the data are indiscriminately viewed by institutions using the data. Moreover, data outsourcing also creates security and privacy concerns because it separates data ownership and data management [4]. Therefore, access control for the data users is necessary to protect personal information and provide only data that meet the attributes of the data user that will use the data. To this end, data access control technology using attribute-based encryption (ABE) [5] has recently attracted attention as a promising technology.

In the case of ciphertext-policy ABE (CP-ABE) [6], each original datum is encrypted in relation to the access control structure set in advance by the encryptor. Data users can only decrypt the ciphertext if the set of attributes he or she uses satisfies the ciphertext access structure. IoT data producers need to be able to provide their data only to organizations that want them through the gateway to ensure privacy. Therefore, since the access structure for IoT data must be determined, using CP-ABE is suitable for the IoT environment.

Additionally, if the cloud server manages the computation and communication of most systems, including outsourced data and access control, it is vulnerable to a single point of failure and data management due to centralization issues [7]. In order to solve this problem, research on the decentralization of cloud servers using blockchains has recently been conducted [8,9]. On the other hand, since IoT data are transmitted and received through open channels, malicious attackers can steal the data to perform attacks such as invasions of privacy, data exfiltration, and data abuse. Therefore, to solve these problems, it is necessary to study the application of ABE and blockchain for data privacy provision and access control. In addition, in order to securely store and provide data, gateways, cloud servers, and data users need to verify that they are valid entities through key agreement.

Therefore, in this paper, we suggest a security system that provides authentication while providing access control. We analyze the trends and problems of systems for secure access control and management of data generated in the IoT environment, and present the direction of blockchain-based access control and key agreement to solve these problems.

The main motivations and contributions of this study based on the problems and challenges mentioned above are as follows:Unlike existing IoT data access control systems using blockchains, the proposed system guarantees data protection through mutual authentication and key agreement. The detailed method is as follows: The proposed system provides mutual authentication based on bilinear pairing and secure key agreement based on the DBDH assumption. In addition, it provides secure data outsourcing and data access control based on CP-ABE by using the session key generated through key agreement.The gateway and the cloud server generate a session key through key agreement and mutual authentication, and the gateway can safely outsource data through the session key. Gateways can also prove data validation through self-signing when uploading data. Data users can request data to the cloud server and verify the received data through the gateway’s signature. Thus, the system can provide data accountability.Since the proposed system utilizes a public permissioned blockchain, only data users, gateways, and cloud servers registered with the *TA* (trusted authority) can use the blockchain as a participant. By auditing the blockchain through data users, nonrepudiation of data can be avoided.Detailed formal security validation utilizing the widely accepted “AVISPA Software Verification Tool” [10], “indistinguishability game against selective chosen plaintext attack (IND-CPA)”, and “informal (nonmathematical) security analysis” shows that the suggested system guarantees safety against multiple potential attacks on smart city environments utilizing IoT.Testbed experiments with cryptographic primitives in a laptop environment were performed using the popular “Multiprecision Integer and Rational Arithmetic Cryptographic Library (MIRACL)” [11].

The remainder of this paper is organized as follows: Section 2 reviews papers on data access control using CP-ABE and blockchain in IoT environments. Section 3 outlines the proposed system model, blockchain, access structure, bilinear pairing, DBDH assumption, and adversary model. Section 4 describes our proposed data access control system. Section 5 describes the results of formal security validation using AVISPA and IND-CPA, and Section 6 describes the results of informal security analysis. We analyze the efficiency and security features of the protocol in Section 7. Finally, Section 8 concludes the paper.

## 2. Related Works

Numerous studies on data access control using CP-ABE have been proposed; its application to the IoT environment has also been proposed. In 2007, Ling and Newport [12] proposed a CP-ABE method that can be applied to both positive and negative attributes using an AND gate access structure. They proposed a structure that has been proven to be secure with plaintext selected under the decisional bilinear Diffie–Hellman (DBDH) assumption. Lewko and Waters [13] suggested a CP-ABE method based on multiauthority, and argued that their system does not require collaboration between rapid institutions. However, in the initialization phase, all agencies must set key parameters, so their structure is impractical for large-scale systems.

In order to efficiently store and manage data, systems in which data are outsourced to a cloud server and controlled have been also proposed [14,15,16,17,18]. Yeh et al. [14] proposed a system that can collect patient information from IoT devices and use it for smart healthcare. For data integrity in their system, data are pre-encrypted before uploading to cloud servers, giving patients access control to data. Miao et al. [15] proposed a CP-ABE-based data access control and keyword search system structure in a cloud-enabled mobile crowdsourcing environment. Liu et al. [16] proposed an e-healthcare record system that uploads and shares health data collected from wearable IoT devices to a cloud server and protects the personal information of patients based on CP-ABE. Ding et al. [17] proposed a structure that can ensure data security in IoT systems by using a pairing-free-based CP-ABE in IoT systems. Lu et al. [18] proposed a secure data sharing system in cloud computing that ensures data privacy protection in resource-constrained mobile terminals. However, since these studies are data access control systems based on cloud servers, a centralization problem may occur, which may cause a single-point-of-failure problem.

Therefore, CP-ABE systems have been proposed for access control of IoT data based on blockchains to solve this centralization problem [19,20,21,22,23,24]. In 2018, Zhang et al. [19] proposed a user-controlled data sharing system with privacy protection using fine-grained access control based on a blockchain and CP-ABE. In 2019, Ding et al. [20] also proposed an ABE access control system for IoT. Blockchain technology was used to record the distribution of properties to prevent single-point errors and data tampering. They demonstrated that authentication can ensure strict access control, but there is no algorithm or protocol for this in their system. Guo et al. [21] suggested a multiauthority blockchain-based ABE protocol for telemedicine systems. Unfortunately, Son et al. [25] figured out that Guo et al.’s protocol [21] is not suitable for real-world environments as patients must maintain their own attribute keys. Yang et al. [22] proposed an EHR sharing system utilizing cloud computing based on ABE and blockchains. In 2021, Wei et al. [23] designed an ABE algorithm for multiauthority scenarios with resource-constrained IoT devices in mind, thereby shifting data management to a blockchain instead of a central server. Qin et al. [24] also proposed a blockchain-based CP-ABE system using cloud computing with consideration to the resource limitations of IoT devices. However, the authentication they proposed [19,20,21,22,23,24] is authentication for data, not mutual authentication between entities participating in communication. For secure communication, the session key must be calculated by performing mutual authentication and key agreement.

Blockchain-based CP-ABE access control systems have been proposed in various smart environments using IoT devices. However, most studies do not provide mutual authentication, key agreement, data access control, validation and accountability at the same time. Therefore, we propose a structure that guarantees a secure data outsourcing process through mutual authentication and key agreement and provides data access control using CP-ABE technology. In addition, our proposed structure proposes an access control system that provides functions of data nonrepudiation, data accountability, and data validation based on a blockchain.

## 3. System Models and Preliminary Work

We present the proposed system model for IoT data access control considering data users in the different IoT environments. We describe blockchain characteristics, ABE, and the adversary model used in our system. Table 1 is an explanation of abbreviations and symbols used in this paper.

### 3.1. System Model

Our proposed data access control system model is described in Figure 1. The proposed system model consists of the following four entities:**Cloud Server (CS)**: A set of CSs forms a “*CS* network”, where a distributed ledger is maintained for block additions. CSs are honest but curious entities. Moreover, the *CS* receives the IoT data and provides the data to the data user when the user’s attribute value matches. In addition, the *CS* uploads data such as data attributes, signature values, and public keys to the blockchain to solve the centralization problem.**Gateway (GW)**: Gateways are distributed in various smart environments that make up the smart city. The gateway collects IoT data from each environment and uploads them to a cloud server with attribute-based encryption appropriate for each attribute.**Data User (DU)**: Data user refers to a person who charges fees using IoT data or provides services such as artificial intelligence, finance, and medical care using IoT data. The data user requests an attribute key from the *TA*. After that, the *TA* can request data matching the attribute from the cloud server, and can obtain the original data by decrypting the received data through the attribute key.**Trusted Authority (TA)**: All data users, gateways, and cloud servers must register with a fully trusted TA.**Blockchain**: In the proposed system model, the blockchain is composed of a public permissioned blockchain. To solve the problem of centralization of CSs, the blockchain stores the storage address of data stored in the *CS*, the public key of each component, hash, data access tree, etc., on behalf of the *CS*. The “practical Byzantine fault tolerance (PBFT) consensus algorithm” [26] has been applied for adding blocks to existing blockchains, verifying blocks, and voting-based consensus algorithms. Data users audit the blockchain ledger. All blockchain members can read the ledger, but only data users and cloud servers can upload transitions to the blockchain. When the *DU* requests information from the *CS*, the *CS* checks whether the access tree of the requested information and the attributes of the *DU* match through the blockchain. If the attributes match the access tree, the *CS* passes the encrypted data to the *DU*.

In the setup phase, TA generates and publishes parameters necessary for the system and tree. During the registration phase, DUs, GWs, and CS are registered with TA through closed channels. Through the attribute key generation phase, DU can ask TA for a key that matches his attribute, and use the acquired attribute key to decrypt the encrypted data. In the authentication phase, GW and CS perform authentication and key agreement for data upload. In the data upload phase, GW uploads data to the cloud server through the agreed session key. Simultaneously, GW uploads the signature as a verification value to verify its own data and the upload time to the blockchain. In addition, CS uploads the attribute tree value of the data and the record address value where the data are stored to the blockchain. In the data request and provide phase, DU requests data from CS, and CS verifies the DU’s request message, checks the attribute value of DU through the blockchain, and transmits the corresponding data to DU. DU downloads the verification value from the blockchain for the transmitted data, verifies that they are valid data, and can decrypt the data with its own attribute key.

### 3.2. Blockchain

A blockchain is a distributed data storage system that can solve the single-point-of-failure problem that can occur by being concentrated in the cloud server. The decentralized nature of blockchains can provide nonrepudiation of data, accountability, and transparency. In addition, the timestamp recorded on the blockchain allows blockchain participants to know the transaction generation time [27]. In general, four types of blockchain platforms are defined:**Public permissionless blockchain**: A public permissionless blockchain provides a ’low trust’ environment where anyone can run nodes and participate in the network. A public permissionless blockchain can be accessed by anyone, and any node can participate in the consensus protocol. Moreover, anyone can read the entire ledger of transactions.**Public permissioned blockchain**: Public permissioned blockchains have rules that determine who can participate in the verification process and launch nodes. They are commonly used by public institutions such as government agencies, businesses, or educational institutions. Whitelisted nodes can participate in the consensus mechanism. Owners create validator nodes that define governance rules for the blockchain, including those who can create new nodes or write to the blockchain. However, read access can be used by anyone who makes the blockchain publicly accessible.**Private permissionless blockchain**: A private permissionless blockchain has no restrictions on who can participate in the consensus mechanism. However, unlike public permissionless convex chains, there are restrictions on who can read and write content on the blockchain.**Private permissioned blockchain**: These blockchains are controlled by a unique group of one or several owners who determine the participants in the consensus mechanism. Only selected user groups can read or write to these blockchains. If public verification of records is not required, consider private permissioned blockchains.

In this paper, only cloud servers and data users of smart cities can write to the blockchain. Therefore, in this paper, a public permission-type blockchain is adopted, and the consensus algorithm uses PBFT.

### 3.3. Access Structure

According to [6], we use the following access tree as an access structure.

Assuming that T is an access tree, T includes $\left(v,num_{v},threshold_{v},par\left(v\right),ind\left(v\right)\right)$, where *v* is a node of T, $threshold_{v}$ is threshold value of *v*, $num_{v}$ is the number of children nodes of *v*, ind(v) is unique index of *v*, and par(v) is a parent node of *v*. Assuming *v* is an internal node, *v* is the threshold gate denoted by AND and OR. AND and OR gates are defined as follows: when $0<threshold_{v}\leq num_{v}$, it is an AND gate if $threshold_{v}=num_{v}$ and an OR gate if $threshold_{v}=1$.

Moreover, in the case where *v* is a leaf node, it is described as the attributes $threshold_{v}=1$. To fit T with attribute set att(v), att(v) have to match the threshold gate at root node τ of T. Here, att(v) is defined only if *v* is a leaf node and represents an attribute related to leaf node *v* in the tree. In the first case, τ is an attribute and its key satisfies the access tree att(v). In the following case, if τ is a threshold gate whose child node is an attribute, the access tree is satisfied when att(v) holds the threshold gate of τ. In other cases, such as where τ is a threshold gate and the child nodes are also threshold gates, the method in the second case can be applied recursively to solve it.

### 3.4. Bilinear Pairing

Let $G_{1}$ and $G_{2}$ be recursive groups with large prime q, and let them be addition and multiplication groups, respectively. Then, a map that satisfies the following conditions can be applied to the bilinear map $e:G_{1}\times G_{1}\in G_{2}$.
**Efficiency**: For all $P,Q\in G_{1}$, e(P,Q) are able to be computed in polynomial time.**Bilinearity**: For all $P,Q\in G_{1}$, and for all $x,y\in Z_{p}^{*}$, e(xP,yQ) is $e{\left(P,Q\right)}^{xy}$.**Nondegeneracy**: Existing $P,Q\in G_{1}$, then $e\left(P,Q\right)\neq 1_{G_{1}}$, where $1_{G_{1}}$ is the identifying element of $G_{1}$.

### 3.5. Decisional Bilinear Diffie–Hellman (DBDH) Assumption

Let $G_{1}$ be a group of order *q*; *P* be a generator of $G_{1}$; and $a,b,c,z\in Z_{q}$ be chosen randomly. The DBDH assumption [28] is that it is difficult for a probabilistic polynomial time adversary A to distinguish $\left(P^{a},P^{b},P^{c},e{\left(P,P\right)}^{abc}\right)$ from $\left(P^{a},P^{b},P^{c},e{\left(P,P\right)}^{z}\right)$. The advantage ε of A is defined as follows:(1)$$\begin{array}{c} |Pr\left[A\left(P^{a},P^{b},P^{c},e{\left(P,P\right)}^{abc}\right)=1\right]-Pr\left[A\left(P^{a},P^{b},P^{c},e{\left(P,P\right)}^{z}\right)=1\right]|\geq \varepsilon \end{array}$$

If there is no A can decide whether $e{\left(P,P\right)}^{z}=e{\left(P,P\right)}^{abc}$, that is deciding whether z=abc or $z\in Z_{q}$, with a non-negligible advantage, the DBDH assumption holds.

### 3.6. Adversary Model

We adopt the widely accepted “Dolev–Yao (DY) threat model” [29] for cryptographic analysis of protocol security. A malicious adversary could, according to the assumptions of the DY model, intercept messages sent over public channels. Attackers can also modify, insert, delete, or eavesdrop on hijacked messages.
An adversary takes full control of transmitted messages sent over an open wireless channel and learns from the messages. The attacker can then modify, remove, or insert a legitimate message.In polynomial time, an adversary is able to only guess one value, because guessing more than one value at a time is “computationally infeasible”, for example, guessing an ID and password at the same time.An adversary can steal or obtain a valid smart card. The adversary can perform a power analysis attack [30,31] on a smart card to steal sensitive information stored on the smart card.

In addition, this paper adopts the assumption of the “CK adversary model” [32], which is a more powerful attack model considering the actual environment. The CK attack model is considered the de facto standard for modeling key exchange protocols. Therefore, in the CK model, for the session key agreed upon between the communicating parties to be secure, the key exchange protocol must minimize the impact of persistent (long-term) or temporary (short-term) secret leaks.

## 4. Proposed Data Access Control System for IoT Environments

In this section, we propose a method of access control for IoT data, which overcomes the limitations and security pitfalls of previous access control methods. In addition, the proposed protocol guarantees stronger security through authentication in the existing access control method.

### 4.1. Setup Phase

*TA* generates public parameters for use in the system’s attribute-based encryption and blockchain. The following steps are followed:

**Step SP1: **TA generates $G_{1}$ and $G_{2}$ in the same order *q*, where $G_{1}$ is an additive cyclic group and $G_{2}$ is a multiplicative cyclic group. Then, TA generates a bilinear map $e:G_{1}\times G_{1}$. TA chooses the secret keys $k_{TA}$ and ζ in $Z_{q}^{*}$, and chooses $P\in G_{1}$, where *P* is a generator. Moreover, TA chooses the hash functions $h_{1}:{\left\{0,1\right\}}^{*}\to Z_{q}$ and $h_{2}:{\left\{0,1\right\}}^{*}\to G_{1}$.

**Step SP2:** TA computes the public key $PK_{TA}=k_{TA}*P$, a factor of an attribute key $F=\frac{P}{k_{TA}}$ and a factor for decryption $e{\left(P,P\right)}^{\zeta }$. Then, TA publishes ($G_{1},G_{2},q,e,P,PK_{TA},F,e{\left(P,P\right)}^{\zeta },$$h_{1},h_{2}$).

### 4.2. Registration Phase

For key agreement and authentication, GW, of the IoT environment, CS and DU have to register at TA. This phase runs through a secure channel.

#### 4.2.1. Cloud Server Registration Phase

This phase is also executed over the secure channel:

**Step CSR1:** A cloud server CS picks its identity $ID_{cs}$ and generates a random number $c_{cs}$. CS computes $PID_{cs}=ID_{cs}⊕c_{cs}$. Then, CS sends $\langle PID_{cs},$$c_{cs}\rangle$ to the trusted authority TA through a closed channel.

**Step CSR2:** After that, TA stores $PID_{cs}$ in a its secure database. TA computes $R_{cs}=h(k_{ta}\left|\right|c_{cs})$ as CS’s private key. After that, TA sends $\langle R_{cs}\rangle$ to CS over a secure channel.

**Step CSR3:** CS computes the public key $PK_{cs}=P*R_{cs}$.

#### 4.2.2. Data User Registration Phase

When a new data user $DU_{i}$ registers with TA, the following steps are followed:

**Step UR1:** $DU_{i}$ chooses unique identity and password $DUID_{i}$ and $DUPW_{i}$. $DU_{i}$ generates random nonces $IU_{i}$ and $a_{i}$, where they are in $Z_{q}^{*}$. Then, $DU_{i}$ computes $HDUID_{i}=h_{1}(DUID_{i}||IU_{i})$ and $HDUPW_{i}=h_{1}(DUID_{i}||IU_{i}||DUPW_{i})$. After that, $DU_{i}$ sends $HDUID_{i}$, $HDUPW_{i}⊕a_{i}$ to TA via closed channels.

**Step UR2:** After TA receives the request message, TA computes $TID_{i}=\left(HDUID_{i}*k_{TA}\right)*PK_{TA}$ and $A_{i}=TID_{i}⊕\left(HDUPW_{i}⊕a_{i}\right)$. TA stores $HDUID_{i}$ with $TID_{i}$ in a its secure memory and stores $A_{i}$ in a smart card SC. Then, TA issues SC to $DU_{i}$. At the same time, TA sends $\langle h_{1}\left(TID_{i}\right),HDUID_{i}\rangle$ to CS via closed channels.

**Step UR3:** After receiving SC, $DU_{i}$ computes $Z_{i}=h_{1}(DUID_{i}||DUPW_{i})⊕IU_{i}$, $B_{i}=A_{i}⊕a_{i}=TID_{i}⊕HDUPW_{i}$, $C_{i}=h_{1}(TID_{i}||HDUPW_{i})$, and $D_{i}=r_{i}⊕HDUPW_{i}$. Then, $DU_{i}$ stores $Z_{i}$, $B_{i}$, $C_{i}$ and $D_{i}$ into SC and computes a public key as $PK_{i}=r_{i}*P$.

**Step UR4:** After receiving message, CS computes $MCS_{i}=h_{1}(HMID_{i}\left|\right|R_{cs})$ and stores $MCS_{i}$ in its secure database. CS also stores $h_{1}\left(TID_{i}\right)$ with $HDUID_{i}$.

#### 4.2.3. Gateway Registration Phase

In this phase, the following steps are performed in the closed channel:

**Step GWR1:** A gateway $GW_{j}$ chooses identity $GID_{j}$ and generates a random nonce $b_{j}$. GW computes $PGID_{j}=GID_{j}⊕b_{j}$. Then, GW generates a public key $PK_{j}=r_{j}*P$ and sends $\langle PGID_{j}\rangle$ to the trusted authority TA via closed channels.

**Step GWR2:** After that, TA computes $TGID_{j}=\left(h_{1}\left(PGID_{j}\right)*k_{TA}\right)*PK_{TA}$ and stores $PGID_{j}$ with $TGID_{j}$ in a its secure database. Then, TA sends $TGID_{j}$ to $GW_{j}$ over a secure channel. At the same time, TA sends $\langle h_{1}\left(PGID_{j}\right),TGID_{j}\rangle$ through secure channels.

**Step GWR3:** CS computes $GCS_{j}=h_{1}(h_{1}(PGID_{j}\left|\right|R_{cs}))$ and stores $GCS_{j}$ in its secure database. CS also stores $TGID_{j}$ with $h_{1}\left(PGID_{j}\right)$.

### 4.3. Attribute Key Generation Phase

In this phase, the data user with attributes $ATTRI_{i}$ requests the attribute key from the TA and provides the corresponding key.

**Step AKG1:** DU chooses his/her attributes $ATTRI_{i}$ and sends it to TA to request the attribute key.

**Step AKG2:** After that, TA generates random nonces $ra_{i},rb_{i}\in Z_{q}^{*}$. In addition, TA computes $At_{i}=F\left(\zeta +ra_{i}\right)$ for all $s\in ATTRI_{i}$, and also computes $At_{i_{s}}=ra_{i}P+rb_{i}H\left(s\right)$ and $At_{i_{s}}^{{}^{\prime }}=rb_{i}P$. Then, TA computes attribute keys $attr_{i}=(At_{i},At_{i_{s}},At_{i_{s}}^{{}^{\prime }}$). Finally, TA sends attribute keys $attr_{i}$ to $DU_{i}$.

**Step AKG3:** After receiving attribute keys, $DU_{i}$ uploads the transaction $Tx_{i}=$$\left(PK_{i},ATTRI_{i}\right)$ to the blockchain.

### 4.4. Authentication and Key Agreement Phase

For uploading the IoT data to the cloud server, $GW_{j}$ and CS authenticate each other. They authenticate each other to secure mutual trust, and later, by establishing the session key SK, $GW_{j}$ and CS can configure a secure communication channel. The detailed steps involved in this step are shown below and are summarized in Figure 2.

**Step AK1:**$GW_{j}$ generates a random number $\beta_{i}$ and timestamp $T_{1}$, and computes $E_{i}=\left(\beta_{i}*r_{j}\right)*P$, $AUTH_{j}=\left(\beta_{i}*r_{j}\right)*PK_{cs}$, $F_{i}=h_{1}\left(PGID_{j}\right)⊕AUTH_{j}$, $MC_{jc}=$$h_{1}(AUTH_{j}$$\left|\right|h_{1}\left(PGID_{j}\right)\left|\right|T_{1})$, and $TAUTH_{j}=TGID_{j}*MC_{jc}$. Then, GW sends a request message $\langle E_{i},$$F_{i},$$PK_{j}$, $TAUHT_{j}$, $T_{1}\rangle$ to CS over an insecure channel.

**Step AK2:** After receiving the message, CS retrieves $h_{1}\left(PGID_{j}\right)$ using $TGID_{j}$ and verifies $h_{1}(h_{1}\left(PGID_{j}\right)\left|\right|R_{cs})\overset{?}{=}GCS_{j}$. If it corrects, CS computes $AUTH_{j}=E_{i}*R_{cs}$, $h_{1}(h_{1}\left(PGID_{j}\right)\left|\right|\beta_{i})=F_{i}⊕AUTH_{j}$, and $MC_{jc}=h_{2}(AUTH_{j}\left|\right|h_{1}\left(PGID_{j}\right)\left|\right|T_{1})$. After that, CS checks $e\left(TAUTH_{j},P\right)\overset{?}{=}e\left(\left(h_{1}\left(PGID_{j}\right)*M_{jc}\right)*PK_{TA},PK_{TA}\right)$. If they are the same, $GW_{j}$ is authenticated. After that, CS generates a $n_{cs}$ and timestamp $T_{2}$. In addition, CS computes $P_{cg}=\left(n_{cs}*R_{cs}\right)*P$, $V_{cg}=\left(n_{cs}*R_{cs}\right)*PK_{j}$, and CS also computes $G_{i}=V_{cg}⊕n_{cs}$, $SK=h_{1}(n_{cs}\left|\right|h_{1}(h_{1}\left(PGID_{j}\right)\left|\right|\beta_{i}))$ as a session key, and $M_{cg}=h_{1}(h_{1}\left(PGID_{j}\right)\left|\right|V_{cg}\left|\right|T_{2})$. Then, CS sends a response message $\langle P_{cg},G_{i},M_{cg},T_{2}\rangle$ to $GW_{j}$ through public channels.

**Step AK3:** After that, $GW_{j}$ checks the validity of $|T_{2}^{\prime }-T_{2}^{\prime \prime }|<\Delta T$. If it is valid, $GW_{j}$ computes $V_{cg}=P_{cg}*r_{j}$ and $n_{cs}=G_{i}⊕V_{cg}$. Then, $GW_{j}$ checks $M_{cg}\overset{?}{=}h_{1}(h_{1}\left(PGID_{j}\right)\left|\right|V_{cg}\left|\right|T_{2})$. If it holds, $GW_{j}$ considers CS as authentic and computes the session key shared with CS as $SK=h_{1}(n_{cs}\left|\right|h_{1}(h_{1}\left(PGID_{j}\right)\left|\right|\beta_{i}))$.

Finally, $GW_{j}$ and CS complete mutual authentication to generate the same session key SK for IoT data upload.

### 4.5. Data Upload Phase

$GW_{j}$ uploads IoT data through the session key agreed with CS. At this time, $GW_{j}$ encrypts data through CP-ABE and uploads them to CS so that only $DU_{i}$ with appropriate attributes can access data sharing. In addition, $GW_{j}$ generates the signature value for data verification of $DU_{i}$. CS stores encrypted data and uploads $GW_{j}$’s signature value, public key, attribute tree, and stored server address value to the blockchain. Detailed steps related to this phase are provided below.

**Step DU1:** $GW_{j}$ chooses an access tree T and root of tree τ. Then, $GW_{j}$ generates a timestamp $TS_{j}$ and selects a random polynomial $q_{\tau }\left(x\right)$ with degree $d_{\tau }=v_{\tau }-1$. $GW_{j}$ generates a random number $x_{j}=q_{\tau }\left(0\right)$ for a leaf node *x* of T. Thereafter, $GW_{j}$ computes $c_{j1}=DATA_{j}*e{\left(P,P\right)}^{\zeta_{x_{j}}}$, $c_{j2}=PK_{TA}*s_{j}$.

For other leaf nodes le of T, $GW_{j}$ chooses a random point $d_{le}$ of polynomial $q_{le}\left(x\right)$. Then, $GW_{j}$ calculates $C_{le}=P*q_{n}\left(0\right)$ and $C_{le}^{\prime }=h_{2}\left(attr\left(le\right)\right)*q_{le}\left(0\right)$ for all leaf nodes le of T. The ciphertext consists of $\delta_{j}=\left(T,c_{j1},c_{j2},C_{le},C_{le}^{{}^{\prime }}\right)$. $GW_{j}$ also computes the signature of data as follows. GW computes $s_{j}=h_{1}(PGID_{j}\left|\right|r_{j}||DATA_{j})$, $S_{j}=s_{j}*P$, and $Sig_{j}=s_{j}+h_{1}(PK_{j}\left|\right|\delta_{j})*r_{j}$ as the signature. Finally, $GW_{j}$ sends $\langle {\left(S_{j},Sig_{j},\delta_{j},TS_{j}\right)}_{SK},h_{1}(PK_{j}\left|\right|\delta_{j}||TS_{j})\rangle$ to CS through a open channel.

**Step DU2:** After that, CS decrypts $\left(Sig_{j},\delta_{j},TS_{j}\right)$ using the session key and checks $h_{1}(PK_{j}\left|\right|\delta_{j}||TS_{j})$. If these values are equal, CS stores $\delta_{j}$ in its database and sets $ADDR_{j}$ to the record address. At the end, CS uploads the transaction $Tx_{j}=$$(S_{j},Sig_{j},PK_{j},T,h_{1}(\delta_{j}||PK_{j}),$$ADDR_{j})$ to the blockchain.

### 4.6. Data Request and Provide Phase

**Step DRP1:** $DU_{i}$ inserts the smartcard SC and inputs $DUID_{i}$ and $DUPW_{i}$. Then, SC computes $IU_{i}^{{}^{\prime }}=Z_{i}⊕h_{1}(DUID_{i}||DUPW_{i})$, $HDUPW_{i}^{{}^{\prime }}=h_{1}(DUID_{i}||IU_{i}^{{}^{\prime }}||DUPW_{i})$, and $TID_{i}^{{}^{\prime }}=B_{i}⊕HDUPW_{i}^{{}^{\prime }}$. SC checks $C_{i}^{{}^{\prime }}\overset{?}{=}h_{1}(TID_{i}^{{}^{\prime }}||HDUPW_{i}^{{}^{\prime }})$. If it is valid, $DU_{i}$ generates random nonce $r_{du}$ and timestamp $TS_{i}$, and computes $r_{i}=D_{i}⊕HDUPW_{i}^{{}^{\prime }}$, $MC_{ic}=h_{1}(AUTH_{i}\left|\right|h_{1}(DUID_{i}||IU_{i}^{{}^{\prime }})||TS_{i})$, $AID_{i}=TID_{i}*MC_{ic}$. After that, $DU_{i}$ obtains the transaction $(Sig_{j},PK_{j},T,h_{1}(\delta_{j}||PK_{j}),ADDR_{j})$. $DU_{i}$ computes $M_{1}=(PK_{i}||ADDR_{j}\left|\right|r_{du}||TS_{i})+r_{i}*PK_{cs}$ and sends the data request message $\langle h_{1}\left(TID_{i}\right)$$AID_{i}$, $PK_{i}$, $TS_{i}$, $M_{1}\rangle$.

**Step DRP2:** After receiving the message, CS retrieves $HDUID_{i}$ using $h_{1}\left(TID_{i}\right)$ and verifies $h_{1}(HDUID_{i}^{{}^{\prime }}\left|\right|R_{cs})\overset{?}{=}MCS_{i}$. If it holds, CS computes $AUTH_{i}=PK_{i}*R_{cs}$ and $MC_{ic}=h_{1}(AUTH_{i}||HDUID_{i}||TS_{i})$. Then, CS checks $e\left(AID_{i},P\right)\overset{?}{=}e\left(\left(HDUID_{i}*MC_{ic}\right)*PK_{TA}\right),PK_{TA})$. If this equality holds, CS obtains $\left(PK_{i},ATTRI_{i}\right)$ from the blockchain. Then, CS computes $(PK_{i}||ADDR_{j}\left|\right|r_{du}||TS_{i})=M_{1}-R_{cs}*PK_{i}$ and confirms that $ATTRI_{i}$ satisfies tree of $\delta_{j}$. If it is met, CS calculates $M_{2}=(\delta_{j}\left|\right|T_{cs})+R_{cs}*PK_{i}$. Then, CS sends the message $\langle M_{2}$$AID_{i}$, $TS_{cs}\rangle$.

**Step DRP3:** After receiving the message, $DU_{i}$ computes $(\delta_{j}||TS_{cs})=M_{2}-r_{i}*PK_{cs}$. Then, $DU_{i}$ checks $h_{1}(\delta_{j}||PK_{j}{)}^{*}\overset{?}{=}h_{1}(\delta_{j}||PK_{j})$ acquired on the blockchain. Depending on the type of root node, data decryption proceeds as follows.
Case 1: If τ is a leaf node, $DU_{i}$ calculates $e(A_{t_{i}},C_{le})$ and $e(A_{t_{i}}^{{}^{\prime }},C_{le}^{{}^{\prime }})$. Then, $DU_{i}$ computes $At_{i_{s}},C_{le}$ and $At_{i_{s}}^{{}^{\prime }},C_{le}^{{}^{\prime }}$. Then, $DU_{i}$ computes $\frac{e(At_{i_{s}},C_{le})}{e(At_{i_{s}}^{{}^{\prime }},C_{le}^{{}^{\prime }})}=e{\left(P,P\right)}^{ra_{i}q_{\tau }\left(0\right)}=K$ for data decryption. Thereafter, $DU_{i}$ can decrypt as follows:
$$\begin{array}{cc} \frac{c_{j1}}{e(c_{j2},At_{i})/K} & =\frac{DATA_{j}*e{\left(P,P\right)}^{\zeta x_{i}}}{e(x_{i}PK_{TA},F\left(\zeta +r_{ai}\right))/K}\\ & =\frac{DATA_{j}*e{\left(P,P\right)}^{\zeta x_{i}}}{e{\left(P,P\right)}^{x_{i}\left(\zeta +r_{ai}\right)}/K}=DATA_{j} \end{array}$$Case 2: We assume that root node τ is a threshold gate and child nodes are attributes. Before we describe the decryption computation, we define the symbols $c_{\tau }$ and $\Delta_{ind\left(le\right),c_{\tau }}\left(x\right)$. $c_{\tau }$ is a set of child nodes of the root node, and $\Delta_{ind\left(le\right),c_{\tau }}\left(x\right)$ is Lagrange coefficient, where $\Delta_{ind\left(le\right),c_{\tau }}\left(x\right)=\Pi_{o\in c_{\tau },ind\left(o\right)\neq ind\left(le\right)}$$\frac{x-ind(o)}{ind(le)-ind(o)}$.$DU_{i}$ computes $\frac{e(At_{i_{s}},C_{le})}{e(At_{i_{s}}^{{}^{\prime }},C_{le}^{{}^{\prime }})}=e{\left(P,P\right)}^{ra_{i}q_{\tau }\left(0\right)}=K_{le}$ for all leaf nodes le. After that, $DU_{i}$ computes decrypt key:
$$\begin{array}{cc} \prod_{le}K_{le}^{{▵}_{ind\left(le\right),c_{\tau }\left(0\right)}} & =\prod_{le}{\left(e{\left(P,P\right)}^{ra_{i}q_{le}\left(0\right)}\right)}^{{▵}_{ind\left(le\right),c_{\tau }\left(0\right)}}\\ & =\prod_{le}{\left(e{\left(P,P\right)}^{ra_{i}q_{\tau }(ind\left(le\right)}\right)}^{{▵}_{ind\left(le\right),c_{\tau }\left(0\right)}}\\ & =e{\left(P,P\right)}^{ra_{i}q_{\tau }\left(0\right)}=K \end{array}$$Then, $DU_{i}$ can decrypt the IoT data.

### 4.7. Data Validation Phase

If the data users want to verify that the gateway information is correct, data verification can be performed during this phase. This data validation ensures that the gateway is accountable for its own data and that the data user can obtain the reliability of the data. A detailed description of this phase is provided bellow:

**Step DVP:**$DU_{i}$ obtains $S_{j}$, $Sig_{j}$, $PK_{j}$, and $h_{1}(\delta_{j}||PK_{j})$ from the transaction related to the data. $DU_{i}$ computes $Sig_{j}*P=s_{j}*P+h_{1}(PK_{j}\left|\right|\delta_{j})*r_{j}*P=S_{j}+h_{1}(PK_{j}\left|\right|\delta_{j})*PK_{j}$. Then, $DU_{i}$ checks $S_{j}=Sig_{j}-h_{1}(PK_{j}\left|\right|\delta_{j})*PK_{j}$. If it is valid, $DU_{i}$ can be considered as data validation completed.

### 4.8. Block Formation and Addition Phase

In the key generation phase and data upload phase, $DU_{i}$ and CS create a transaction and upload it to the blockchain. We describe it in detail in terms of CS in this section, and the block construction and addition of $DU_{i}$ is similar. The “practical Byzantine fault tolerance (PBFT) consensus algorithm” [26] has been applied for adding blocks to existing blockchains, verifying blocks, and voting-based consensus algorithms. The block structure is depicted in Figure 3, and the entire algorithm of block addition is given in Algorithm 1.
**Algorithm 1** PBFT Consensus for Block Addition in Blockchain by Cloud Server  1:**Input:** Block $(Block_{m}$ as shown in Figure 3, transactions pool $\left(Tx_{p}ol\right)$, transactions threshold $(Tx_{thresh=t}$, number of CS nodes: $n_{cs}$, minimal approval $\left(Min_{approve}=2*\left(n_{cs}-1\right)/3+1\right)$  2:**Output:** Commitment for block addition (CMP)  3:Assume that a cloud server node ($CS_{l}$) is elected as a leader  4:$CS_{l}$ picks a fresh timestamp and creates a block $Block_{m}$ with $Tx_{pool}$  5:$CS_{l}$ sets CMP=NULL and sends $Block_{m}$ to follower cloud server nodes ($CS_{k}$$\left(k\neq l|k=1,2,\ldots ,n_{cs}\right)$) for voting request  6:**for** each follower $CS_{j}$ **do**  7:   **if** (($Tx_{j}=valid$) and (MR=valid) and ($ECDSA.sig_{Tx}=valid$) and (CBHash=valid)) **then**  8:     Set CMP=CMP+1  9:   **end if** 10:**end for** 11:**if** ($CMP\geq Min_{approve}$) **then** 12:   Add $Block_{m}$ to the blockchain 13:   Broadcast commitment message to CS 14:**end if**

#### 4.8.1. Block Formation Phase

At the data upload phase of our system, the data generated by $GW_{j}$ are uploaded to CS using SK agreed between $GW_{j}$ and CS at the authentication and key agreement phase. CS safely gathers *t* counts of data, filters that information, and then generates *t* counts of transactions $Tx_{j}=$$(S_{j},Sig_{j},PK_{j},T,h_{1}(\delta_{j}||PK_{j}),ADDR_{j})$, for j= 1, 2,…, t, to contribute to the transactions pool. To describe this in detail in terms of the data upload phase, CS computes the Merkle tree root (MR) for transactions $Tx_{j}$ and calculates “elliptic curve digital signature” for transactions $Tx_{j}$ as $ECDSA.sig_{Tx}=ECDSA.sig_{gen}\left(Tx_{msg}\right)$, where $Tx_{msg}=h_{1}(Tx_{1}||Tx_{2}||\ldots ||Tx_{j}||PK_{cs}\left||MR\right)$.

#### 4.8.2. Block Addition Phase

After block formation phase, the MR for the transaction existing in the block is verified. In addition, CS conducts a voting-based PBFT consensus algorithm. The CS nodes $CS_{l}|l=1,2,\ldots ,n_{cs}$ ($n_{cs}$ represent the number of peers in CS) form a distributed P2P network. Here, each CS node is considered a peer node that is responsible for adding blocks. After the CS peer node receives the $Block_{m}$, peer node verifies it with the existing transaction pool. When all transactions in $Block_{m}$ are confirmed by the transaction pool, the peer puts a valid vote into the commit message pool. CS constantly checks the commit message pool and checks when the minimum approval $\left(Min_{approve}\right)$ for block additions on the blockchain is reached; where $Min_{approve}=2*\left(n_{cs}-1\right)/3+1$, the new block $Block_{m}$ will be added to the blockchain.

## 5. Formal Security Validation: AVISPA Simulation Study and IND-CPA

In this section, we utilize the “AVISPA simulation tool” [10] and IND-CPA to verify the security of the proposed system.

### 5.1. AVISPA Simulation

We use the “AVISPA Simulation Tool” [10] in this section to validate our proposed system security against man-in-the-middle and replay attacks.

In AVISPA, there are four backends: “tree automata based on automatic approximations for analysis of security protocols (TA4SP)”, the “SAT-based model checker (SATMC)”, the “on-the-fly-mode-checker (OFMC)” and the “constraint-logic-based attack Searcher” (CL-AtSe)”. Among these, the SATMC and TA4SP backends can not aid the “bitwise exclusive OR (XOR)”. However, since our system has an XOR operation, two backends are not suitable for analysis. Therefore, we adopt two backends, OFMC and CL-AtSe, which support XOR operation, and use them for analysis. In the proposed system, “High-Level Protocol Specification Language (HLPSL)”, a language supported by AVISPA, is used to implement the basic roles of CS and $GW_{j}$. Figure 4 shows the HLPSL implementation of the role user.

At transition 1, GW sends the request message $\{PGID_{j}$} to TA using SND operation and SKgwta, which means the secure channel. The declaration $secret(\{Bj^{{}^{\prime }},$Rj},sp3,{GW}) means that the random nonce $B_{j}$ and secret key $R_{j}$ are only known to GW.

At transition 2, GW receives the $TGID_{j}$ from TA. In login and authentication phase, GW sends the message $\left\{E_{i},F_{i},TGID_{j},PK_{j},TAUTH_{j},T_{1}\right\}$ to CS through insecure channel. The declaration $witness(GW,CS,gw\_cs\_bei,Bei^{\prime })$ means that GW generates a random nonce $\beta_{i}$ for CS.

At transition 3, GW receives the message $\left\{P_{cg},G_{i},M_{cg},T_{2}\right\}$ from CS. The declaration request(CS,GW,cs_gw_ncs,$Ncs^{\prime })$ specifies that CS request to the GW for checking the value of $n_{cs}$.

HLPSL of cloud server is implemented similarly to gateway’s HLPSL. In addition, it implements “composite roles and goals for sessions and environment” of the proposed system through HLPSL. AVISPA used in this section is a security validation simulation based on the DY model [30]. Figure 5 gives the analysis results performed on the CL-ATse and OFMC backends. The figure clearly shows that the proposed system can be resistant to “replay and man-in-the-middle attacks”.

### 5.2. IND-CPA Security

We prove the confidentiality property of our system with the game of IND-CPA. In our scheme, the game is defined as follows.
**Init.** The adversary A gives a challenge access structure $T^{*}$.**Setup.** The simulator X executes Setup phase and sends the public parameters to the adversary A.**Phase 1.** A queries multiple private keys corresponding to $q_{1}$ different sets of attributes $\left(ATTRI_{1},\ldots ,ATTRI_{q_{1}}\right)$ where $ATTRI_{i}\notin T^{*}$.**Challenge.** A submits two plaintext $DATA_{0}$ and $DATA_{1}$, where $|DATA_{0}|=|DATA_{1}|$ to the simulator X with $T^{*}$. X flips the coin b∈{0,1}, encrypts $DATA_{b}$ under $T^{*}$, and sends the ciphertext $CT^{*}$ to A.**Phase 2.** A repeats Phase 1 with the attribute sets $\left(ATTRI_{q_{1}+1},\ldots ,ATTRI_{q}\right)$ where $ATTRI_{i}\notin T^{*}$.**Guess.** A outputs a guess $b^{\prime }$ of *b* to the simulator X. If $b^{\prime }=b$, A wins the game.

The adversary A’s advantage ε in this game is defined as $\varepsilon =|Pr\left[b^{\prime }=b\right]-\frac{1}{2}|$. If A in probabilistic polynomial time can be played with a non-negligible advantage ε, then we prove that the problem of the DBDH assumption can be solved with ε/2.

**Proof.** Assume that the adversary A wants to take advantage of ε to subvert the system. We build a X simulator to play the DBDH game with a ε/2 advantage. We proceed through the simulation process as follows. The B challenger randomly picks $a,b,c,z\in Z_{q}$ and generator $P\in G_{1}$. B flips a coin to obtain a random value μ∈{0,1}. If μ=1, $Z=e{\left(P,P\right)}^{z}$, which means $\left(P^{a},P^{b},P^{c},e{\left(P,P\right)}^{z}\right)$. Otherwise, $Z=e{\left(P,P\right)}^{abc}$ means $\left(P^{a},P^{b},P^{c},e{\left(P,P\right)}^{abc}\right)$. After that, B transmits the results to X.**Init.** The simulator X runs A to create access structure $T^{*}$ that A hopes to attack. Then, A transmits it to X.**Setup.** X computes public parameters $\left\{PK_{TA}=k_{TA}*P,F=\frac{P}{k_{TA}},e{\left(P,P\right)}^{\zeta }\right\}$, where ζ=ab. Then, X sends them to A.**Phase 1.** A requests multiple private keys $\left(attr_{i1},\ldots ,attr_{iq_{1}}\right)$ corresponding to $q_{1}$ different sets of attributes $\left(ATTRI_{1},\ldots ,ATTRI_{q_{1}}\right)$ where $ATTRI_{i}\notin T^{*}$. The simulator X generates random nonces $ra_{i},rb_{i}\in Z_{q}^{*}$. X computes $At_{i}=F\left(\zeta +ra_{i}\right)$ for all $s\in ATTRI_{i}$, $AT_{i_{s}}=ra_{i}P+rb_{i}H\left(s\right)$, $At_{i_{s}}^{{}^{\prime }}=rb_{i}P$, $attr_{i}=\left(At_{i},At_{i_{s}},At_{i_{s}}^{{}^{\prime }}\right)$. Then, X sends $attr_{i}$ to A.**Challenge.** A submits $T^{*}$ to the X simulator with plain text $DATA_{0}$ and $DATA_{1}$ of equal length. X randomly tosses a coin to obtain b∈{0,1}. If μ=0, then $Z=e{\left(P,P\right)}^{abc}$. In this case, we let $x_{j}=c$, then $e{\left(P,P\right)}^{abc}=e{\left(P,P\right)}^{\zeta x_{j}}$ and $c_{j1}=DATA_{b}*e{\left(P,P\right)}^{abc}$. Otherwise, if μ=1, then $Z=e{\left(P,P\right)}^{z}$ and $c_{j1}=DATA_{b}*e{\left(P,P\right)}^{z}$. X computes $c_{j2}=PK_{TA}*s_{j}$. Then, X chooses a random point $d_{l_{e}}$ of polynomial $q_{le}\left(x\right)$ and computes $C_{le}=P*q_{n}\left(0\right)$, $C_{le}^{\prime }=h_{2}\left(attr\left(le\right)\right)*q_{le}\left(0\right)$ for all leaf nodes le of T. Then, X sends $\delta_{j}=\left(c_{j1},c_{j2},C_{le},C_{le}^{\prime }\right)$ to A.**Phase 2.** The adversary A repeats **Phase 1** to obtain the private keys that are associated with attribute sets $\forall ATTRI_{i}{|}_{q_{1}+1\leq i\leq q}$ and $ATTRI_{i}\notin T^{*}$.**Guess.** A guesses $b^{\prime }$ of *b*. If $b\neq b^{\prime }$, X gives a result 1, otherwise, it gives a result 0. If X gives a result 0, then $Z=e{\left(P,P\right)}^{abc}$. A can obtain practical ciphertext $\delta_{j}$. The advantage in this case is ε, so we obtain $Pr\left[b^{\prime }=b|Z=e{\left(P,P\right)}^{abc}\right]=\frac{1}{2}+\varepsilon$. When X gives a result 0, it means $Z=e{\left(P,P\right)}^{z}$. A obtains the wrong ciphertext, and does not have the advantage of guessing the correct $b^{\prime }$, so it is able to obtain $Pr\left[b\neq b|Z=e{\left(P,P\right)}^{z}\right]=\frac{1}{2}$. Therefore, the probability Pr of a successful game is
(2)$$\begin{array}{cc} Pr & =\frac{1}{2}Pr\left[A\left(P,P^{a},P^{b},P^{c},e{\left(P,P\right)}^{abc}\right)=1\right]\\ \; & \;+\frac{1}{2}Pr\left[A\left(P,P^{a},P^{b},P^{c},e{\left(P,P\right)}^{z}\right)=1\right]-\frac{1}{2}\\ \; & =\frac{1}{2}Pr\left[b^{\prime }=b|Z={\left(P,P\right)}^{abc}\right]\\ \; & \;+\frac{1}{2}Pr\left[b^{\prime }\neq b|Z=e{\left(P,P\right)}^{z}\right]-\frac{1}{2}\\ \; & =\frac{1}{2}\times \left(\frac{1}{2}+\varepsilon \right)+\frac{1}{2}\times \frac{1}{2}-\frac{1}{2}\\ \; & =\frac{\varepsilon }{2} \end{array}$$Therefore, our scheme ensures IND-CPA security.□

## 6. Informal Security Analysis

We provide an nonmathematical (informal) security analysis of whether the proposed system can provide various security features and safety against possible attacks.

### 6.1. Correctness of Data Decryption Key

If the leaf node le is a root node τ, we check the correctness of the decryption key as follows:$$\begin{array}{cc} \frac{e(At_{i_{s}},C_{le})}{e(At_{i_{s}^{{}^{\prime }}},C_{le}^{{}^{\prime }})} & =\frac{e(ra_{i}P+rb_{i}h_{2}\left(attr\left(\tau \right)\right),q_{\tau }\left(0\right)P)}{e(rb_{i}P,h_{2}\left(attr\left(\tau \right)\right)q_{\tau }\left(0\right))}\\ & =\frac{e\left(ra_{i}P,q_{\tau }\left(0\right)P\right)e\left(rb_{i}h_{2}\left(attr\left(\tau \right)\right),q_{\tau }\left(0\right)P\right)}{e(rb_{i}P,h_{2}\left(attr\left(\tau \right)\right)q_{\tau }\left(0\right))}\\ & =\frac{e{\left(P,P\right)}^{ra_{i}q_{\tau }\left(0\right)}e{\left(h_{2}\left(attr\left(\tau \right)\right),P\right)}^{rb_{i}q_{\tau }\left(0\right)}}{e{\left(P,h_{2}\left(attr\left(\tau \right)\right)\right)}^{rb_{i}q_{\tau }\left(0\right)}}\\ & =e{\left(P,P\right)}^{ra_{i}q_{\tau }\left(0\right)}=K \end{array}$$

### 6.2. Guessing Attacks

The malicious adversary A cannot guess the data user’s $DUID_{i}$ and $DUPW_{i}$ in the proposed system. A obtains the credentials $\left\{Z_{i},B_{i},C_{i},D_{i}\right\}$ stored on the smart card. However, since $\left\{Z_{i},B_{i},C_{i}\right\}$ is encrypted with random numbers $IU_{i}$ and $a_{i}$, A cannot obtain sensitive information. Furthermore, these values are protected via “a one-way collision-free hash function h(·)”. In addition, $D_{i}$ is masked by the unknown parameter $HDUPW_{i}$ and secret key $r_{i}$. As a result, our proposed system can resist guessing attacks.

### 6.3. Tracing Attacks and Provides Anonymity

The adversary A is trying to obtain the real IDs of $DU_{i}$ and $GW_{j}$ to perform a tracking attack. In our system, the user’s real identity $DUID_{i}$ is hidden by $HDUID_{i}$ masked with a random number $IU_{i}$. In addition, the $DU_{i}$ sends the message through the public channel using the temporary ID $TID_{i}$ received from TA via an insecure channel. Moreover, $GW_{j}$ hides its real ID $GID_{j}$ as $PGID_{j}$. $GW_{j}$ sends a message through the public channel with the temporary ID $TGID_{j}$ obtained from TA. So, A cannot know original IDs $DUID_{i}$ and $GID_{j}$. This demonstrates that our system provides anonymity and can resist tracing attacks.

### 6.4. Impersonation Attacks

A may attempt to impersonate each entity by calculating legitimate messages to obtain information. In our system, messages sent over public channels are encrypted using random numbers $\beta_{i}$, $n_{cs}$, $x_{i}$, and $r_{du}$ and secret values $r_{j}$ and $R_{cs}$. Moreover, in the data upload phase, the message is encrypted by the session key SK. A tries to take out these values, but this cannot be carried out. In addition, each of the entities check $e\left(TAUTH_{j},P\right)\overset{?}{=}e\left(\left(h_{1}\left(PGID_{j}\right)*M_{jc}\right)*PK_{TA},PK_{TA}\right)$, $M_{cg}\overset{?}{=}h_{1}(h_{1}\left(PGID_{j}\right)\left|\right|V_{cg}\left|\right|T_{2})$, and $e\left(AID_{i},P\right)\overset{?}{=}e\left(\left(HMID_{i}*MC_{ic}\right)*PK_{TA}\right),PK_{TA})$. Therefore, the proposed system can provide protection against impersonation attacks.

### 6.5. Ephemeral Secret Leakage Attacks

In the authentication and key agreement phase, $GW_{j}$ and CS establish the session key $SK=h_{1}(n_{cs}\left|\right|h_{1}(h_{1}\left(PGID_{j}\right)\left|\right|\beta_{i}))=h_{1}(n_{cs}\left|\right|h_{1}(h_{1}\left(GID_{j}⊕b_{j}\right)\left|\right|\beta_{i}))$ in our system. The SK depends on “ephemeral secrets $n_{cs}$ and $\beta_{i}$” and long-term secret $b_{j}$. Even if the attacker “short-term secret $n_{cs}$ and $\beta_{i}$” is compromised for A, guessing SK without long-term secret $b_{j}$ is “a computationally difficult problem.” Likewise, even if “long-term secret $b_{j}$” is compromised to A, deriving SK is also “computationally difficult. except for short-term secrets. Since SK between the gateway and the cloud server is distinct and unique, leaking SK from a session to A is “computationally infeasible” as it applies both short-term and long-term secrets without having to compute another session key in another session. Therefore, the proposed system prevents ephemeral secret leakage attacks.

### 6.6. Mutual Authentication and Key Agreement

At our system, GW and CS use the $TAUTH_{j}$ and $M_{cg}$ values to authenticate each other by verifying the message. Every transmitted message is changed with a random number and current timestamps. GW and CS authenticate each other through an authentication and key agreement phase and compute the same session key SK only if the authentication is complete. Therefore, our system provides key agreement through mutual authentication.

### 6.7. Data Access Control, Validation and Accountability

The proposed system can provide access control to IoT data of $GW_{j}$. $GW_{j}$ establishes an access tree for IoT data and uses it to encrypt data and upload them to CS. Then, only $DU_{i}$ with the appropriate set of attributes in the IoT data’s access tree is able to request data from CS and decrypt them with the attribute key. In addition, $GW_{j}$ uploads the signature value of its own data to the transaction on the blockchain. $DU_{i}$ can confirm that the data are uploaded by $GW_{j}$ through the signature value of the transaction, which means that $GW_{j}$ guarantees accountability for its own data when uploading. Thus, the system can provide data access control, validation, and accountability.

## 7. Efficiency Features and Security Analysis

The proposed system is compared with existing competitive data access control systems in the smart city area, such as smart health and smart homes [18,24]. The compared schemes are all schemes using attribute-based encryption. We compare different data access control schemes with each other in terms of communication and communication costs, function, and security features.

### 7.1. Testbed Experiment Using MIRACL

In this section, we apply MIRACL to show an an environment for practical perspective experiments. The MIRACL testbed experiment shows the computation costs of the proposed system. We performed a testbed experiment with cryptographic primitives using the popular “MIRACL” [11] in a laptop environment. Here are the detailed performance details of the laptops we used: “Ubuntu 18.04.4 LTS with memory 8 GiB, processor: Intel Core i7-4790 @ 3.60 GHz × 4, CPU Architecture: 64-bit”. The experiments were run 100 times to determine the time to run “bilinear pairing operation $\left(T_{pair}\right)$”, “ECC signature operation $\left(T_{sig}\right)$, “ECC scalar point multiplication $\left(T_{mul}\right)$”, “ECC point addition $\left(T_{add}\right)$”, “modular exponentiation operation $\left(T_{exp}\right)$”, “map-to-point-hash-function $\left(T_{mtp}\right)$”, “encryption function $\left(T_{enc}\right)$”, “decryption function $\left(T_{dec}\right)$”, and “one-way-hash-function $\left(T_{h}\right)$”. Thereafter, the average execution time in milliseconds for these functions or operations over 100 run was recorded: 6.587 ms, 0.546 ms, 2.547 ms, 0.013 ms, 0.164 ms, 7.564 ms, 0.001 ms, 0.001 ms, and 0.003 ms, respectively.

### 7.2. Security and Function Feature Comparison

This section presents the results of comparison of the proposed system with related existing approaches in terms of security and functionality. Table 2 presents the results of the comparison. Previous studies do not provide data accountability, nor do they provide the functions of mutual authentication and key agreement, whereas the proposed method meets all essential security and functional requirements for data access control in a smart city environment.

### 7.3. Computation Cost Comparison Analysis

Computational costs are compared, taking into account the data upload and data request and provide phases, and follow the testbed experiment results reported in Section 7.1.

We use the average time required on the platform for the data owner/gateway/IoT device, cloud server, and data user costs, respectively. Table 3 shows the comparison results of the computation costs. In Table 3, *n* means the number of attributes. We assumed that *n* is 5 to obtain the total computation costs. It can be observed that the total computational costs of our system are slightly higher than those of the other systems. The proposed system uses traditional CP-ABE, which has proven safety rather than efficiency. Moreover, as shown in Table 2, the proposed system can provide mutual authentication, key agreement, and data accountability that other systems cannot provide, and it is safe against attacks from various security aspects.

### 7.4. Communication Cost Comparison Analysis

For comparison analysis of the communication costs during the data upload and data request and provide phases between the proposed system and other systems, the *l* column matrix, encryption data, hash function output value (using SHA-256), public key, identity, ECC value, chain code, index, and timestamp are taken as 32l bits, 256 bits, 256 bits, 256 bits, 160 bits, 256 bits, 256 bits, 256 bits, and 32 bits, respectively.

Table 4 indicates that our system requires communication costs of 2112 bits to exchange three messages for data upload and data download. On the other hand, the schemes of Lu et al. [18] and Qin et al. [24] require communication costs of 32l+ 1952 bits for three messages and 2208 bits for three messages.

## 8. Conclusions

In this paper, we proposed an access control system for IoT data in various IoT environments based on CP-ABE and blockchains. Existing systems do not provide mutual authentication and key agreement for secure communication. However, the proposed system guarantees secure communication through these two properties. In addition, the proposed system can provide data validation and accountability to data users. To verify the safety of our system, formal and unofficial security analysis was performed, and the proposed system was compared and analyzed with existing systems in terms of security and functionality. Through the analysis results, it was found that the proposed system is safe against guessing, tracing, ESL, and session key disclosure attacks, unlike existing systems. In addition, our protocol can be said to be an efficient protocol because it has a computation cost similar to or lower than that of existing systems and a lower communication cost than existing systems.

In the future, we plan to design a more efficient access control system. In this paper, we used the traditional CP-ABE, but we need to design an efficient ABE for a more efficient system design. In traditional CP-ABE, when the number of users or the number of attributes increase, the number of pairing operations increases. This will increase the computational cost of the system, which will make it impossible to provide real-time services to users in the IoT environment. In order to solve this problem, there is a need to study a new method of access control in the future. If we develop an efficient access control method even if the number of users and attributes increases, we will be able to design an access control system that is more suitable for the IoT environment.

## Figures and Tables

**Figure 1 sensors-23-05173-f001:**
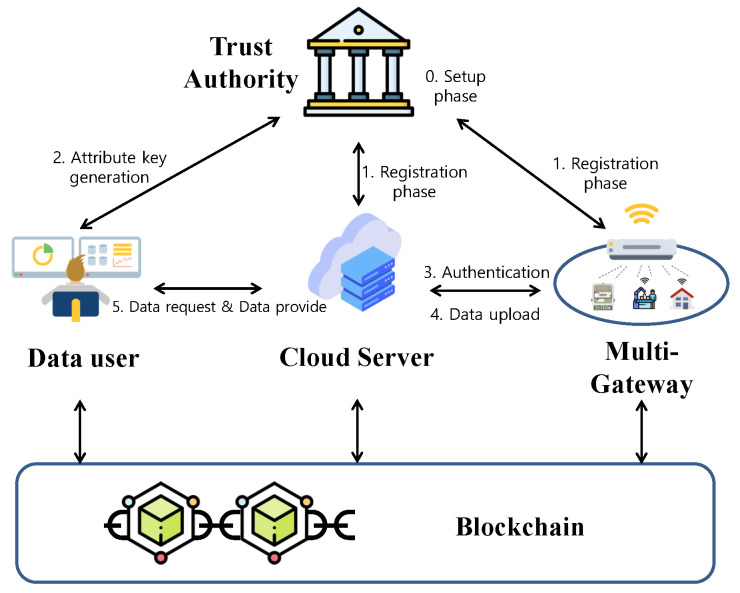
Proposed system model (author’s own processing).

**Figure 2 sensors-23-05173-f002:**
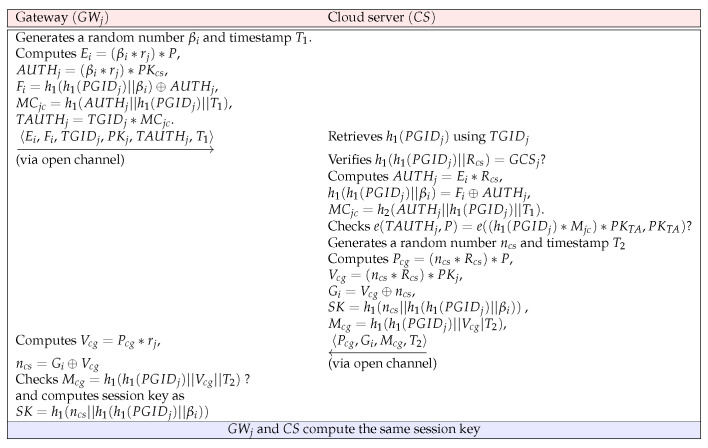
Authentication and key agreement phase (author’s own processing).

**Figure 3 sensors-23-05173-f003:**
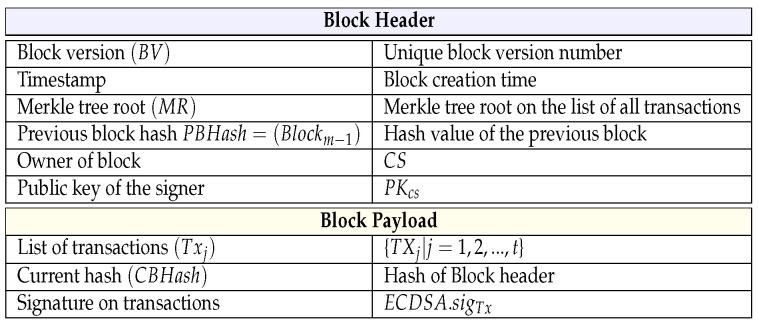
Formation of a block on the transactions by *CS* (author’s own processing).

**Figure 4 sensors-23-05173-f004:**
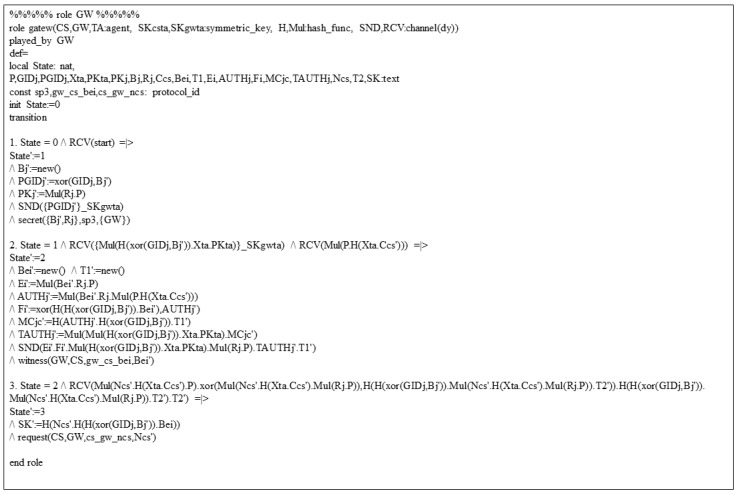
HLPSL specification for user (Author’s own processing).

**Figure 5 sensors-23-05173-f005:**
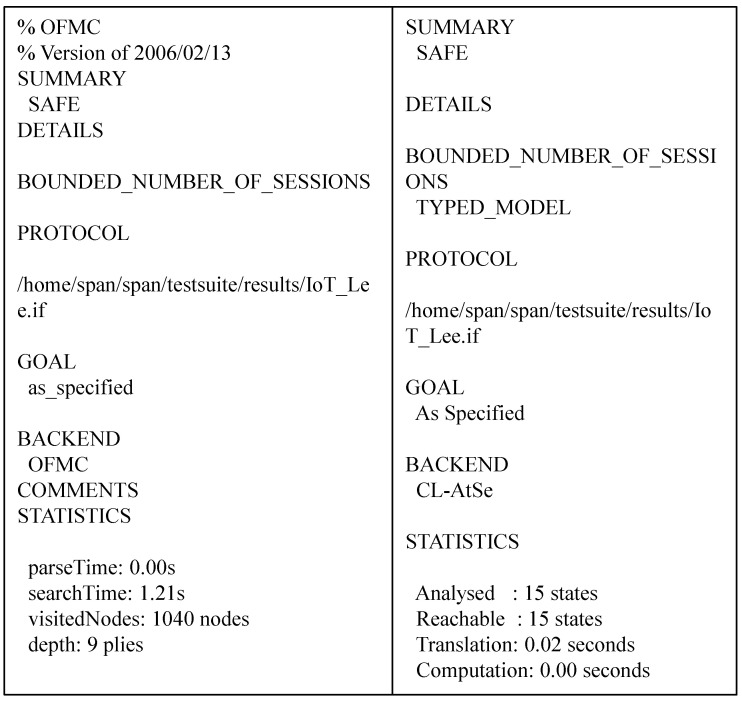
Simulation results on OFMC and CL-AtSe.

**Table 1 sensors-23-05173-t001:** Notations (author’s own processing).

Notations & Abbreviations	Meanings
IoT	Internet of Things
ABE	Attribute-based encryption
CP-ABE	Ciphertext-policy ABE
DBDH	Decisional bilinear Diffie-Hellman
$DU_{i}$, $DUID_{i}$, $DUPW_{i}$	*i*th data user
	and his/her identity and password, respectively
$GW_{j}$, $GID_{i}$	*j*th gateway and its identity, respectively
CS, $ID_{cs}$	Cloud server and its identity, respectively
$HDUID_{i},PGID_{j},PID_{cs}$	The hidden identity of data user,
	gateway, and cloud server, respectively
TA	Trusted authority
$R_{cs}$, $r_{i}$, $r_{j}$, $k_{TA}$	The secret key of CS, $DU_{i}$, $GW_{j}$, and TA, respectively
$PK_{cs}$, $PK_{i}$, $PK_{j}$, $PK_{TA}$	The public key of CS, $DU_{i}$, $GW_{j}$, and TA, respectively
$ATTRI_{i}$	The attribute of $DU_{i}$
$attr_{i}$	The attribute private key of $DU_{i}$
T, τ	Access tree and root of tree
SK	The session key established among $GW_{j}$ and CS
*K*	The data decryption key
$h_{1},h_{2}$	Hash function and map-to-point hash function
||	Data concatenation operator
⊕	Bitwise exclusive-or operator

**Table 2 sensors-23-05173-t002:** Security and function properties comparison (Author’s own processing).

Security and Function Properties	Lu et al. [18]	Qin et al. [24]	Proposed
$SF_{1}$	x	-	o
$SF_{2}$	x	-	o
$SF_{3}$	o	o	o
$SF_{4}$	x	o	o
$SF_{5}$	-	-	o
$SF_{6}$	-	-	o
$SF_{7}$	x	x	o
$SF_{8}$	o	o	o
$SF_{9}$	x	x	o

o: provide the security property x: does not provide the security property -: does not consider $SF_{1}$: Guessing attack $SF_{2}$: Anonymity and tracing attacks $SF_{3}$: Replay and man-in-the-middle attacks $SF_{4}$: Impersonation attack $SF_{5}$: ESL attack $SF_{6}$: Session key disclosure attack $SF_{7}$: Mutual authentication and key agreement $SF_{8}$: Data validation $SF_{9}$: Data accountability.

**Table 3 sensors-23-05173-t003:** Computation costs comparison (Author’s own processing).

System	Data Owner/Gateway	Cloud Server	Data User	Total Costs
Lu et al. [18]	$\left(1+n\right)T_{mul}+2T_{exp}+T_{h}$	$T_{h}+T_{sig}+\left(n\right)T_{exp}$	$T_{pair}+2T_{mul}$	110.307 ms
	≈15.613 ms	$+\left(3+n\right)T_{mul}+\left(2n\right)T_{pair}\approx 83.013$ ms	≈11.681 ms	
Qin et al. [24]	$5T_{mul}+9T_{exp}+T_{add}+4T_{h}$	$\left(8+6n\right)T_{mul}+6T_{pair}$	$T_{exp}+2T_{mul}$	156.802 ms
	≈14.236 ms	≈137.308 ms	≈5.258 ms	
Proposed	$\left(4+2n\right)T_{mul}+\left(n\right)T_{mtp}+2T_{h}+T_{pair}$	$3T_{h}+2T_{mul}+2T_{add}+T_{pair}$	$5T_{mul}+\left(n\right)T_{pair}+\left(n\right)T_{exp}$	138.305 ms
	$+T_{add}+T_{enc}\approx 80.081$ ms	≈11.716 ms	$+6T_{h}\approx 46.508$ ms	

*n*: number of attribute (assuming that *n* = 5).

**Table 4 sensors-23-05173-t004:** Communication costs comparison (Author’s own processing).

System	Number of Messages	Number of Bits
Lu et al. [18]	3	32l + 1952
Qin et al. [24]	3	2208
Proposed	3	2112

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
