# Peer review of "Blockchain-Based Data Access Control and Key Agreement System in IoT Environment"

_sensors, 2023, doi:10.3390/s23115173_

Round 1

Reviewer 1 Report

There is a lot of work invested in the article and the information is really valuable but shoud be solved some issues before to be published (I think that in the next revision it will not have any problem to be published if you take into account the recommendations below, please - thank you)

-          Usually keywords don't take (over) sequences from the title (eg. blockchain, data access control and so on) - please replace them in the way to reflect the article ideas and not just be redundant

-          Please specify the source of each figure / table (e.g. “Author's own processing” or other expressions / sources, if the case) between square or round brackets after the name of the table. Improve each figure / table (which is not “self- processing) with your own contribution

-          References 21, 25, 26, 27, 28, 37, 38, 39 and 41are self-references, please? Besides this, not all sources are correctly cited (eg. souce 40, 41– and so on -  refer to sources that are not found in the article’s text). Please remove absolutely all self-citations from the text and add text for 40 and 41 references. The elimination of self citations also presupposed the "reformulation" of the paragraph in the article - corresponding to the quoted source

-          Mir&Sch.2022!

-          In order not to abound with abbreviations and explanations in abstract and article’s body, I recommend placing the explanation of all / each character(s) or abbreviation(s) (e.g. for each parameter, variable, attribute of and so on – and define each formula like  equation, lema, theorem, proof of theorem and so on.The article must be easy to understand, both for specialists and for those less familiar with the subject. Please check the consistency and accuracy of each of them

-          The section of introduction should include (even briefly at the end of the chapter): the context of the study, which are the main results presented  in short, which is the originality of this paper, the main implication policy of these results and a description of the structure of the paper - the role of each section of the paper. Some of them are missing - please fill it accordingly

-          The “Literature Review”, partially assimilated in your article with Related works chapter, should include in more detail the “gap” in existing literature / studies and the innovative aspects brought by this paper (analysis for existing “literature” and the novelty and originality brought by this paper should be highlighted regarding previous studies) - please detail the gaps in the existing literature (partially done in different chapters, including Related works & System Models and Preliminary chapters) and state more clearly / more explicitly the manner in which the article addresses these gaps.

-          Please mentione (after Conclusions chapter) more clear (the subjective and) limiting nature of the study (the limits of the research and the way in which these limits will be addressed in the future – if will be) and argue opinion regarding a possible modification of the investigation indicators also to reflecte and to have a holistic view on the topic

Just small typos

Reviewer 2 Report

1.     In the abstract, results are written in general. Results are written in general. Outcomes must be in terms of some metrics.

2.     Novelty, keywords, and technical approach are to be stated in the abstract.

3.     In the introduction, the research questions, and implications descriptions are more adequate; but the methods and results are still briefer and less detailed.

4.     The first contribution should be omitted since this is not by just this paper.

5.     I don't understand how a factor of an attribute key is calculated. Whether the expression is wrong

6.     In 4.1. Setup Phase h1 : {0, 1}^--> Z_q, why GCSj = h1(h1(PGIDj||Rcs)).

7.     roi is not explained.

8.     Is Algorithm 1 self-designed?

9.     SF8 is not a security property. It is recommended that compare security properties only in Table 2 and explain the compared schemes briefly.

10.  Discussion in Experimental needs to strengthen.

11.  Strong conclusion required. It does not reflect your contribution in terms of quantified outcomes.

Extensive editing of English language required

Reviewer 3 Report

The authors propose a blockchain-envisioned data access control system in IoT enabled smart city environment that can provide secure data outsourcing, access control, data non-repudiation, data accountability and data validation. My main concerts are listed below:

1- The conclusion section has to be improved with a deeper analysis.

2- More technical papers about IoT and blockchain

[1] A Decentralized Location-Based Reputation Management System in the IoT Using Blockchain. IEEE Internet of Things Journal, 9(16), 15100-15115.

[2] Internet of things. Manual of digital earth, 387-423.

Round 2

Reviewer 1 Report

Dear Respectable Authors

As I told you last time, there is a lot of work invested in the article and the information is really valuable but should be solved some issues before to be published

I had the request to remove all self-citations from the text: 21, 25, 26, 27, 28, 37, 38, 39, 41. and so on. I checked and noticed that you replaced self-citations from the text - without interfering with the text.

The elimination of citations also presupposed the "reformulation" of the paragraph in the article - corresponding to the quoted source. Otherwise, either the initially cited source "was not" correlated with the paragraph in the article, or the source cited later "is not" correlated with the paragraph in the article

I will give you some examples of paragraphs for which you have modified the (number of) source, but you have not changed anything in the text (are identical or very similar):

 ‘Unfortunately, their protocol is not suitable for real-world environments as patients must maintain their own attribute keys [21]. Yang et al. [22]” (V1) versus “Unfortunately, their protocol is not suitable for real-world environments as patients must maintain their own attribute keys [22]. Yang et al. [23]” (V2)

“For secure communication, the session key must be calculated by performing mutual authentication and key agreement [2528]” (V1) versus “For secure communication, the session key must be calculated by performing mutual authentication and key agreement [26,27]” (V2)

 Fig. 4 gives the analysis results performed on the CL-ATse and OFMC backends. The figure clearly shows that the proposed system can be resistant to “replay and man-in-the-middle attacks”. [3739]” (V1) versus “Fig. 5 gives the analysis results performed on the CL-ATse and OFMC backends. The figure clearly shows that the proposed system can be resistant to “replay and man-in-the-middle attacks”. [36,37]” (V2)

 And so on…..

I identified the fact that you replaced the self-citations from 21, 25, 26, 27, 28, 37, 38, 39, 41 (and so on)  with the self-citations from 22, 26, 27, 36, 37 (and so on) and I understand the fact that you did not want to remove the self-citations at all - you just changed their number.

Having said that, I want to inform you that absolutely I do not want to adopt a negative decision regarding the revised document, but that I am very close to reject the article in the situation you continue to treat the reviewer's recommendations in the same way or to decline my competence so, that another reviewer can make the decision in my place - because the approach seems to me (I'm not sure and that's why I don't want to express myself in this sense) tends to become a problem of professional ethics.

The choice is yours.

Until then, I will ask you to revise the text again, in the hope that you will identify the right formula to be published (as I also wish)

 Besides this:

 -          Please specify the source of each figure / table (e.g. “Author's own processing” or other expressions / sources, if the case) between square or round brackets after the name of the table. Improve each figure / table (which is not “self- processing) with your own contribution - I asked you before, but……

-          Please mentione (in a pharagraph after Conclusions chapter) more clear (the subjective and) limiting nature of the study (the limits, lacks, weaknesses of the research and the way in which these limits will be addressed in the future – if will be) and argue opinion regarding a possible modification of the investigation indicators also to reflecte and to have a holistic view on the topic - I asked you before, but……

Thank you so much all of you in advance.

Only very small typos

Round 3

Reviewer 1 Report

Dear Respectable Authors

As I told you previous time/s (two times actualy), there is a lot of work invested in the article and the information is really valuable but must be solved some issues before to be published

I had the request to remove all self-citations from text and noticed that you replaced some self-citations from text - without interfering with the text in some of them.

As I told you last time, the elimination of citations also presupposed the "reformulation" of the paragraph in the article - corresponding to the quoted source. Otherwise, either the initially cited source "was not" correlated with the paragraph in the article, or the source cited later "is not" correlated with the paragraph in the article

I will give you (again) some examples of paragraphs for which you have modified the (number of) source, but you have not changed anything in the text (are identical or very similar):

“For secure communication, the session key must be calculated by performing mutual authentication and key agreement [26,27]” (V2) versus “For secure communication, the session key must be calculated by performing mutual authentication and key agreement [?,?]” (V3)

 “Fig. 5 gives the analysis results performed on the CL-ATse and OFMC backends. The figure clearly shows that the proposed system can be resistant to “replay and man-in-the-middle attacks”. [36,37]” (V2) versus “Fig. 5 gives the analysis results performed on the CL-ATse and OFMC backends. The figure clearly shows that the proposed system can be resistant to “replay and man-in-the-middle attacks. [? ?]” (V3)

 Besides this:

“Unfortunately, their protocol is not suitable for real-world environments as patients must maintain their own attribute keys [22]” (V2) versus Unfortunately, Son et al. [22] figured out 128 that Guo et al.’s protocol is not suitable for real-world environments as patients must maintain their own attribute keys” (V3) should reference Guo et al., not Son et al – or both of them. 

I will choose "Accept after minor revision (corrections to minor methodological errors and text editing)" option, but I will also select "Yes" to the question "Did you detect inappropriate self-citations by authors?" Next, it depends on the editors if they will consider appropriate to publish the article before or after you make the requested changes (if you make them)

Thank you all so much for colaboration.
